# Validating the Perceived Active School Travel Enablers and Barriers–Parent (PASTEB–P) Questionnaire to Support Intervention Programming and Research

**DOI:** 10.3390/ijerph20105874

**Published:** 2023-05-19

**Authors:** Andrew F. Clark, Melissa Thomas, Adrian Buttazzoni, Matthew Adams, Stephanie E. Coen, Jamie Seabrook, Danielle Tobin, Trish Tucker, Jason Gilliland

**Affiliations:** 1Human Environments Analysis Laboratory, Department of Geography and Environment, Western University, London, ON N6A 3K7, Canadaabuttazz@uwo.ca (A.B.);; 2Department of Geography and Environment, Western University, London, ON N6A 3K7, Canada; 3Department of Geography, Geomatics and Environment, University of Toronto—Mississauga, Mississauga, ON L5L 1C6, Canada; 4School of Geography, University of Nottingham, Nottingham NG7 2RD, UK; 5School of Food and Nutritional Sciences, Brescia University College, Western University, London, ON N6A 3K7, Canada; 6School of Occupational Therapy, Western University, London, ON N6A 3K7, Canada; 7Children’s Health Research Institute, London, ON N6C 2V5, Canada; 8Lawson Health Research Institute, London, ON N6A 4V2, Canada; 9School of Health Studies, Western University, London, ON N6A 3K7, Canada; 10Department of Paediatrics, Western University, London, ON N6A 3K7, Canada; 11Department of Epidemiology & Biostatistics, Western University, London, ON N6A 3K7, Canada

**Keywords:** active school travel, children’s health, interventions, parental perceptions, physical activity, questionnaire, school travel planning, validation

## Abstract

A child’s ability to participate in active school travel (AST) is complicated by several factors. Of particular note are parental controls, which are informed by their perceptions of the local built and social environments, assessments of their child’s skills, and convenience preferences, among other considerations. However, there is currently a lack of AST-specific scales that include validated parental perception measures related to such notable barriers and enablers, or those that tend to frame their AST decision-making processes. Framed within the social-ecological model of health behaviour, the aims of the present paper were thus threefold, specifically to (1) outline and test the construct validity of measures delineating parental perceptions of barriers and enablers to AST, (2) evaluate the reliability and consistency of the developed measures, and (3) connect these measures to develop broader constructs for use in the Perceived Active School Travel Enablers and Barriers–Parent (PASTEB–P) questionnaire. To achieve these aims, a mixed-methods approach featuring cognitive interviews and surveys, along with qualitative (thematic analysis) and quantitative (Cohen’s Kappa, McDonald’s Omega, and confirmatory factor analysis) analyses, was applied across two studies. The validation processes of the two studies resulted in the development of fifteen items comprising seven distinct constructs (barriers: AST Skills, Convenience, Road Safety, Social Safety, and Equipment Storage; enablers: Supportive Environment and Safe Environment) related to parental perceptions of AST. The developed PASTEB–P questionnaire can be used to inform and evaluate AST intervention programming and can be applied for AST research purposes.

## 1. Introduction

Increased childhood physical inactivity and sedentary behaviour are growing public health concerns in many countries across the globe [1,2]. Although physical activity (PA) plays a fundamental role in overall child health and well-being, rates of compliance with PA guidelines among younger populations are decidedly lacking; for example, 81% of adolescents aged 11–17 years were insufficiently active as of a 2016 global assessment [3]. These developments are especially notable in developed countries like Canada, where only 9.3% of children aged 5–17 years meet their daily PA recommendations of 60 minutes or more of moderate-to-vigorous PA (MVPA) [4,5]. Active school travel (AST), or any mode of human-powered, non-motorised transportation (e.g., walking, biking, and rolling) to/from school or the bus stop [6,7], represents both an economically and environmentally sustainable approach to addressing these child health issues as well as a relatively accessible source of PA for many children. Increasing children’s engagement in AST can have substantial physical, developmental, and social benefits for children, including reducing stress [8] and depression and anxiety symptoms [9], as well as improving mental health [10,11], fitness levels [12], cardiorespiratory fitness [13], academic performance [9,10,14], and social cohesion [15]. However, children’s participation in AST is complicated by several factors, including their own knowledge and skills [16], perceptions of community safety, urban design quality [17], and parental controls and perceptions [18]. The latter—parental perceptions of AST—can be an especially determinative influence, as parents’ perceptions of AST as a worthwhile or meaningful activity can both encourage [19] or deter [20] their children’s engagement based on a variety of considerations. Despite the relative importance of parental perceptions of their children’s AST participation, repeated studies have noted methodological concerns regarding the validity of such measures [21,22]. For instance, it has been argued that there is a need to further examine factors that parents perceive as facilitating or obstructing AST [23]. Noting these shortcomings, the aims of this paper are specifically to: (i) develop reliable and validated measures delineating parental perceptions of barriers and enablers to AST; and (ii) connect these validated measures for the purposes of developing broader constructs for use in the Perceived Active School Travel Enablers and Barriers–Parent (PASTEB–P) questionnaire.

### 1.1. AST Barriers and Enablers, School Travel Planning

Despite the well-known benefits of AST in relation to children’s health and well-being, rates of engagement have significantly declined in recent decades [24,25,26]. From a socio-ecological perspective, a child’s decision and ability to participate in AST can be influenced by a myriad of factors, including individual, interpersonal, and environmental factors [22,27,28,29,30]. Of relevance to the present paper are those factors related to parent and child demographics, perceptions, and attitudes that can limit or encourage AST [22,31]. Notably, such factors that have been reputed to impact perceptions of and participation in AST include socioeconomic status [32,33,34], ‘stranger danger’ [35], driving behaviours [36], crime [37], availability of travel companions [38], pedestrian infrastructure quality, land use [39,40,41], commute distance [42,43], and local transportation policy [44]. 

To address the many reported perceived barriers that can limit AST and, conversely, promote the behaviour, one common programme strategy that has been applied in various contexts is the development and implementation of STP interventions. STP is a multi-component, school-based AST intervention designed to encourage AST via a committee comprised of some combination of a lead enabler, public health practitioners and/or nurses, school principals and staff, student representatives, parents, police, crossing guards, and traffic engineers [45,46]. Through a five-step process (chronologically ordered: set-up, baseline data collection, action plan development, implementation, and evaluation), interventions can employ a number of different programmatic strategies, including educational campaigns (e.g., skill development classes), enforcement policies (e.g., police ticketing policies), encouragement or promotion strategies (e.g., ‘walk to school days’), engineering projects (e.g., installation of signage), and equity initiatives (e.g., programming for targeted groups) [47]. While several reviews have evaluated both the implementation strategies and effectiveness of STP interventions [29,48,49], related AST research has expressed concerns regarding the methodological robustness of such evaluations and specifically the quality of such scholarship as it relates to the validity of AST perception measures [21,22]. Therefore, there is a need to better explore the factors that parents perceive as facilitating or obstructing AST and to adequately conceptualise these factors [23]. Moreover, among the current AST-specific scales examining parental perceptions of AST topics, such scales have focused primarily on parental intentions and behavioural beliefs [50] or have lacked the inclusion of a full scope of socio-barrier items [51] in their tools. To support efforts aimed at evaluating school- or community-level AST interventions and developing a tool containing a comprehensive suite of social and physical environmental measures, we frame our scale development process within socio-ecological theory.

### 1.2. Theoretical Background (Social-Ecological Theory)

To frame the development of our scale, we used social-ecological theory [52]. More precisely, we selected the social-ecological model of health behaviour to inform the development of our scale. Importantly, the social-ecological model of health recognises that factors at multiple levels can influence individual health-related behaviours and experiences [53] and is thus an ideal approach to understanding and examining AST perceptions. As noted above, AST is a complex behaviour influenced by factors. These various influences may be conceptually understood to exist at different and distinct levels of the social-ecological model, specifically the intrapersonal, interpersonal, environmental, and policy levels [29,52]. Within the social-ecological model, noteworthy potential influences on parental perceptions of AST could include: one’s vitals and demographics (e.g., gender and age), perceptions, and attitudes [28,31] (intrapersonal); socio-economic status [54], family behaviours and beliefs [55], perceptions of neighbourhood safety [56], and fears of strangers [54] and crime [37] (interpersonal); street connectivity, residential density, and mixed-land-use [57,58] (environmental); and municipal planning and school board policies [59] (policy). Previously, the theory’s extensive scope and suitability with respect to addressing both a variety of AST topics and informing different groups (e.g., individuals, school communities, and policymakers) have not only seen it used widely in primary research regarding AST but also as a desirable model to inform AST intervention design [29,52]. Noting this spread in AST scholarship, in the present paper we apply the social-ecological model to inform the structure of our scale delineating parental AST-specific perceptions. Specifically, the social-ecological model was used to ensure the comprehensiveness (i.e., considering individual, interpersonal, community, and policy influences) and relevance (i.e., targeting integral items) of the scale in service of making it applicable to AST scholars, local stakeholders, and community collaborators in their varied AST pursuits (e.g., primary research, intervention design).

### 1.3. Study Aims and Objectives

Due to the uncertainties pertaining to the validity of AST perception constructs and measures, this study aims to assess the construct validity and internal consistency of AST-specific measures reflecting barriers and enablers to participation from the perspective of parents. To this end, our study has three specific objectives: To outline and test the face and construct validity of measures delineating parental perceptions of barriers and enablers to AST.To evaluate the reliability and consistency of the measures reflected in the various parentally perceived barriers and enablers to AST; andTo connect these validated measures for the purposes of developing broader constructs for use in the Perceived Active School Travel Enablers and Barriers–Parent (PASTEB–P) questionnaire.

In developing the PASTEB–P questionnaire, the objective of the present paper is to support AST intervention research, such as that related to the STP programme, by providing a set of validated and reliable measures of an array of AST-specific parental perceptions that capture constructs across the socio-ecological landscape.

## 2. Methods

This study draws from two pilot studies that were conducted between May 2021 and March 2022. Both studies received ethics approval from Western University’s Non-Medical Research Ethics Board (NMREB#: 118382) and the research offices of a regional school board and were guided by the social-ecological model in their designs (e.g., questions developed and evaluated). The first pilot study, referred to as the Validation Pilot Study, was launched in May 2021 and completed in early February 2022. This pilot study aimed to outline and inform the validity of multiple AST perception measures through cognitive interviews, which were completed concurrently with surveys. The second pilot study, referred to as the Test-Retest Pilot Study, was launched in January 2022 and finished in early March 2022. It asked parents/guardians to complete the survey two times, 5–7 days apart, to determine the reliability of the perception measure included in the survey. The participants for both pilot studies were adults (e.g., parent/guardian, grandparent) with a child who engages in, to some extent, AST (i.e., regular or occasional AST families; see Table 1). Children from these parent/guardian dyads also participated in a separate arm of the study; this paper only reports results related to the parents/guardians.

### 2.1. Recruitment and Sample

#### 2.1.1. Validation Pilot Study Protocol

Recruitment for the Validation Pilot Study was conducted using quota sampling methods to obtain 75–100 parent-child dyads. The aim was to have a minimum of 10 participating dyads representing different target demographics, including children in grades 4–8 (i.e., 9–14 years) and of different neighbourhood-level socioeconomic status (SES). Participants were recruited through three different mediums: social media (via Twitter and Facebook ads), schools (via principals and send-home letters), and community recreational and public health facilities (via posters and other recruitment materials). Interested study participants were directed to visit a website where they were provided more information about the study and instructed to provide their contact information and answer eligibility questions. All eligible participants were contacted by the project coordinator and scheduled for an online cognitive interview on Zoom. Prior to starting the interviews, parents/guardians were instructed to review the letter of information (LOI), ask any questions, and sign the letter of consent. The final sample for this study included 80 parent-child dyads representing 24 schools from across Ontario, with most located in Southwestern Ontario (see Table 2). 

#### 2.1.2. Test-Retest Pilot Study Protocol

Recruitment for the Test-Retest Pilot Study was conducted using a similar quota sampling approach that sought to enrol participating dyads representing the same target demographics: dyads featuring children in grades 4–8 (i.e., 9–14 years) and of diverse neighbourhood-level SES backgrounds. Recruitment for this study was conducted through a social media campaign on Twitter and Facebook, where ads were used to direct interested study participants to visit an enrollment website. All eligible participants were contacted by the project coordinator and scheduled to attend our online survey sessions on Zoom, during which groups of parents/guardians completed the survey with a research assistant who provided help and instruction. Prior to completing the survey, parents/guardians were instructed to review the LOI, ask any questions they had, and sign the letter of consent. The final sample included 86 parents/guardians and 152 children from 73 schools across the Canadian province of Ontario that were also mostly based in the Southwestern region (see Table 2).

### 2.2. Data Collection

Both studies gathered information documenting participant demographics, travel behaviours, independent mobility attitudes, and perceived barriers and enablers of AST. Importantly, the Validation Pilot Study had parents/guardians assess the barriers and enablers with respect to their family, while the Test-Retest Pilot Study had parents/guardians assess the barriers and enablers with respect to individual children. Table 3 presents the initial set of constructs and their component items that were developed for both studies through our team’s examination of relevant studies evaluating barriers and enablers to AST, which included a review of over 100 academic articles, e.g., [22,23,45]. Participant responses were sorted depending on child school bus eligibility (i.e., lives within walking distance vs. outside walking distance [1.6 km]), for which specific prompts were used: **Prompt for parents/guardians of children who live within walking distance**: ‘I do not feel comfortable letting my child walk, bike, or roll to school because…’**Prompt for parents/guardians of children eligible for the school bus**: ‘I do not feel comfortable letting my child walk to and from the bus stop because…’

For both perceived parental/guardian barriers and enablers of AST, participants in both studies were asked to rank their agreement with each statement on the following four-point Likert scale: strongly agree, agree, disagree, and strongly disagree. A neutral category was not included in the Likert scale used in our studies, as neutral categories are known to make it difficult to differentiate between true neutral responses and non-responses [60,61]. In line with the social-ecological framing of the study, questions were developed to inquire about perceived barriers and enablers that were directly or indirectly related to AST influences across all levels of the behavioural model (i.e., intrapersonal, interpersonal, environmental, and policy).

#### 2.2.1. Validation Pilot Survey and Cognitive Interview Study Protocol

In the Validation Pilot Study, participants first met with the research team on Zoom to discuss the protocol and then subsequently completed their surveys and cognitive interviews via a Qualtrics link. The cognitive interview approach of this study necessitated a think-aloud protocol where participants interacted with a research assistant who observed the survey completion and checked in with them to ensure their progress as well as capture their thoughts regarding the survey. On average, the cognitive interviews took approximately 25–30 min to complete. After the surveys and interviews were finished, each participant was asked to engage in a quick 1-on-1 debriefing assessment. While debriefing, participants were asked to elaborate on any questions or comments they had regarding the survey. This process provided an opportunity for the research assistant to probe participants for more details on items that they had difficulty interpreting or to flesh out any lingering or partially articulated ideas or comments that had not been fully followed up on in the cognitive interviews. Over the course of this study, 25 participants did not complete the survey and/or interview process.

#### 2.2.2. Test-Retest Pilot Survey Study Protocol

In the Test-Retest Pilot Study, participants were asked to complete the family survey two times within a week (5–7 days between surveys). In accordance with the study’s protocol, participants were required to schedule a meeting with the research team via Zoom to receive survey instructions, provide consent, and then receive a link to the online survey, where they would arrange a future time to meet with a research assistant who would administer the survey and remain present to ensure its completion. At the conclusion of the first survey, parents/guardians were asked to provide a date within 5–7 days when they could complete the same survey a second time. The same protocol was followed for the second survey. 

### 2.3. Data Analysis 

A mixed-methods analytical approach was used in this study based on the specific aims that corresponded with each arm (i.e., study) outlined in the previously described study protocol and data collection sections. Component parts of this paper’s analysis were: (1) qualitative analyses of the Validation Study survey and cognitive interviews; (2) Kappa Tests to examine repeatability and reliability of the Test-Retest Study survey answers; and (3) McDonald’s Omega with confirmatory factor analysis (CFA) to examine the convergent and discriminant validity of the constructs of barriers and enablers to AST from the Test-Retest Study surveys. Additional details pertaining to each analytical method are provided below.

#### 2.3.1. Qualitative Analysis

To address objective 1 (outline and test the face and construct validity of parental/guardian perceptions), data from the Validation Pilot Study’s survey and cognitive interviews were qualitatively analysed. Each completed survey—those composed of the initial questions that were developed via the research team’s literature review searches—was accompanied by a cognitive interview with individual participants. All interviews, along with the notes from the ensuing debrief conversations, were transcribed verbatim using Microsoft Stream, verified by the research team for their accuracy, and then analysed using an inductive thematic analysis approach [62]. As a part of the inductive analysis process, two research team members first independently sorted and grouped repeating ideas, concepts, and sentiments from the participant interviews. These initial codes containing the repeating ideas, concepts, etc. were then merged to develop larger, more descriptive themes [62] regarding each barrier and enabler construct and its component questions/items. Coding discrepancies between the research team members were brought to the larger group and discussed as needed until a consensus agreement was reached regarding the point in question. Eventually, the thematic analysis resulted in the reframing of specific questions that contributed to each barrier and enabler construct, as well as the addition or removal of specific items that were deemed unsuitable for use (see the right column in Table 3). All qualitative analyses were completed in Taguette, an open-source qualitative research software [63].

#### 2.3.2. Cohen’s Kappa Test

To address objective 2 (evaluate reliability of perception measures), Cohen’s Kappa tests were calculated to determine the level of intra-rater (i.e., within individual) reliability across two time points with the Test-Retest Pilot Study Protocol data (see Section 2.2.2 for full details). The Kappa test was conducted for the barriers and enablers of the test and retest phases using two versions of the data: (1) the original 4-point Likert questions described in Table 3 were assessed with Kappa tests to develop more comprehensive scales for AST research (i.e., higher agreement required); and (2) each of the individual items within the different barrier and enabler constructs were then recoded into binary responses (i.e., agree vs. disagree) and analysed with Kappa tests to offer simpler and more efficient scales that could be used for intervention development purposes (i.e., lower agreement required). Cohen’s Kappa coefficient is appropriate for qualitative data, including nominal and ordinal variables [64]. Kappa results were assessed based on previously established cut points, where ≥0.81 was interpreted as ‘almost perfect’ agreement, 0.61–0.80 as ‘substantial’ agreement, and 0.41–0.60 as ‘moderate’ agreement [65,66,67]. This study follows guidelines provided by past work [68,69,70] that only Kappa values above 0.8 provide sufficient evidence of strong agreement and can be used for research evaluating how AST is influenced by individual barriers and enablers to AST. Barriers and enablers with lower Kappa values between 0.41 and 0.80 are not as reliable in their agreement and will only be recommended to be used to inform intervention development. All statistical analyses were completed in SPSS (IBM, version 25, Markham, ON, Canada).

#### 2.3.3. McDonald’s Omega with Confirmatory Factor Analysis (CFA)

For objectives 2 and 3 (i.e., the final validation of perception measures into larger constructs for use in the PASTEB–P), McDonald’s Omega and CFAs were run with the Test-Rest Pilot Survey data (i.e., four-point Likert scale values (strongly disagree (1); disagree (2); agree (3); strongly agree (4)). We elected to use McDonald’s Omega as opposed to Cronbach’s Alpha’s, as the Omega’s possess better and more realistic data assumptions [68], notably that they are less restrictive in their assumptions and allow the means and variance of true scores, as well as the error variance, to vary, which is unlike the assumptions of Alpha’s [69]. McDonald’s Omega tests were run to assess the internal consistency of each construct, with interpretive thresholds being roughly equivalent to Alpha’s, specifically that Omegas between 0.50 and 0.69 were interpreted as ‘sufficient’ or ‘acceptable,’ and correlations above 0.70 were interpreted as ‘good’ or better [70]. 

Two CFAs were run to verify the construct validity of the perceived AST construct—one for the latent AST barrier constructs and one for the latent AST enabler constructs. CFAs allow for the authentication of the factor structure of a set of observed variables through examining the relationships between observed variables (items) and their underlying latent variables (constructs) [71]. Factor loading cut points followed previously established guidelines, which have outlined 0.32 (poor), 0.45 (fair), 0.55 (good), 0.63 (very good), or 0.71 (excellent) as recognised interpretative cut points [72]. Any factor loadings <0.32 will therefore be removed from the final questionnaire. Model fitness for the CFAs was assessed using Chi-square (CMIN), Chi-square divided by Degree of Freedom (CMIN/DF), and root mean square error of approximation (RMSEA). CMIN values, which indicate the amount of difference between the sample and fitted covariance matrices, should ideally be >0.05 (if significant, the model can be considered unsatisfactory) [73]. CMIN/DF values, meanwhile, should ideally be ≤3 to indicate acceptable fit; however, values of ≤5 have been suggested to represent reasonable fit [74]. Last, RMSEA values, which indicate the difference between the observed covariance matrix per degree of freedom and the predicted covariance matrix, suggest acceptable fit from 0.10–0.05 and excellent fit at <0.05 [75,76]. All CFAs were completed in the AMOS (IBM, version 24, Markham, ON, Canada) software package for SPSS.

## 3. Results 

Full sample demographics for both studies are presented in Table 2. Both samples were comprised of participants who were predominantly born in Canada and relatively well-educated. While the samples were reflective of a roughly 50/50 split between boys and girls among the children being reported on, the parents/guardians in the samples were heavily skewed towards women. As per the sampling protocol, both samples included a minimum of 10 participating dyads representing the different target demographics of children in grades 4–8 (ages 9–14) and neighbourhood-level SES. Per parental/guardian reports (see Table 1), children living within walking distance (i.e., generally identified in Ontario as ≤1.6 km between home and school) made up just over 50% of the Validation Pilot Study sample and just under 50% of the Test-Retest Study sample. Active commuters represented a plurality of participants in both samples.

### 3.1. Face Validity (Qualitative Analysis)

Themes developed because of the inductive analysis provided insight into how the measures of perceived AST barriers and enablers and their supporting questions could be reworked to more comprehensively include the sentiments, views, and preambles/framing structures that parents/guardians deemed lacking from the initial survey outline developed by the research team. To this end, four key themes were developed: response options, definitions, question framing, and ambiguity regarding mode of travel questions. Summaries of the eventual revisions that were applied to the surveys based on this feedback are detailed in the discussions below relating to each theme. However, complete revisions can be found in Table 3, which notes both new (i.e., ‘ADDED’) and old (i.e., ‘REMOVED’) questions that the two surveys contained. Quotations are used to provide more depth to or emphasise the points being discussed.

#### 3.1.1. Response Options

Although more technical in nature, feedback regarding response options was frequent and mostly focused on issues related to the options themselves being too limited. Parents/guardians often recommended the inclusion of a ‘neutral’ response option in many sections of the various constructs of the survey, with a statement elaborating, “I do not know. Can they consider it somewhere in the middle? For some of the questions, I am not 100%. I do not agree, but I also do not disagree. Therefore, it is hard to pick between agree or disagree,” and “There needs to be a ‘neutral’, ‘I do not know’, or ‘not applicable’ [option on the survey].” On these recommendations, a ‘not applicable’ option was added to certain relevant questions, accompanied by a preamble with instructions for participants; however, as noted earlier, ‘neutral’ options were not included on the surveys for reasons related to concerns around discerning a ‘true neutral’ from indifference in participant responses.

#### 3.1.2. Definitions

Several AST-related terms were reported to be unclear or too vague, which hindered participants’ comprehension of specific barrier and enabler questions. Of note, seemingly relative terms like ‘walking distance’ (a ‘convenience’ barrier) tended to generate confusion. In response to these types of concerns, a number of specific items outlining such terms of concern were removed (e.g., for ‘convenience’—“It is too cold in the winter” and “It is too hot in the spring/fall’ to commute) from the revised surveys to improve clarity. Other, more specific terms like ‘maintained’ (a ‘road safety’ barrier) were likewise cited as terms that were too relative and thus heavily subject to personal judgements. To redress this issue, as alluded to above, a ‘not applicable’ response option was added to the revised surveys on certain highlighted questions. Other examples of identified terms of concern included ‘other adult’ and ‘role’, for which key terms and examples were added to the revised survey to clarify the usage and meaning of such terms. 

#### 3.1.3. Question Framing

Several participants reported confusion regarding the way certain questions were framed, leading to interpretation difficulties. Specific framing devices like ‘I do not feel comfortable’ were the source of much uncertainty, with one participant explaining, “I thought some of the wording was confusing for me because it was asking me about why I felt uncomfortable. It was a sort of double negative”, and another added “It is almost like double negative; [this might] impede your ability or your child’s ability”. This sentiment carried over to a few enabler questions, notably pedestrian presence, with other parents/guardians remarking, “I am a little bit confused that people inside the homes can see pedestrians and cyclists” and “If people were walking or cyclists riding their bikes, they could be seen by people in their homes along the way. If I am looking out my front window or back window, could I see people walking or cycling?” Across these examples, the essence of such framing issues was concisely summarised by one parent, who elaborated on the lack of specific or guiding instructions:

I would say some of the wording, like just the interpretation, I could never do without asking a researcher some direct questions about what they were getting at, so I would not say it was easy. I would say it was difficult. I would say some of the questions were not correctly phrased. They were not really specific enough for me to know what you were trying to get at.

Ultimately, the revised surveys included expanded instructions for all questions of concern, specifically removed instances of double negatives, and added clear prefaces for more abstract questions/phrases (e.g., the ‘safe environments’ question about the presence of pedestrians). 

#### 3.1.4. Ambiguity Regarding Mode of Travel Questions

Similar to the preceding theme, participant responses also commonly highlighted a need for more clarity pertaining to questions inquiring about the different modes of travel being examined. Here concerns were primarily raised regarding what constitutes the nature of constructs like ‘supportive environments’ enablers (e.g., seeing trees on a bus route vs. walking question) and what exactly ‘walking distance’ barriers (e.g., does walking distance include the journey to a bus stop question) specifically entailed. To address these issues, the revised surveys were structured to expressly ask parents/guardians of children who walked to the bus stop, as well as those who were bussed to school, to skip the barrier questions, as this survey item sought to capture barriers to AST with school as the destination.

### 3.2. Item Reliability

Table 4 presents the results of the Cohen’s Kappa test analysis, which examined the agreement in parental/guardian responses regarding the measures reflecting various perceived barriers to AST. Barrier measures were organised into the five constructs outlined in the revised survey of Table 3 (i.e., ‘skills’, ‘convenience’, ‘road safety’, ‘social safety’, and ‘equipment storage’). In the first analysis of Kappa test statistics according to the 4-point Likert, responses indicated ‘moderate’ or poorer (<60%) agreement with respect to all 21 specific barrier items. However, in the second analysis, where the items were recoded into binary variables of ‘agree’ or ‘disagree’, the Kappa test scores increased considerably. In particular, 11 of the 21 barrier items were of ‘substantial’ agreement (60–80%), of which one (‘no peers to walk with’) was of ‘almost perfect’ (≥81%) agreement. Conversely, six items were of ‘moderate’ (41–60%) agreement.

Results of the analysis examining the agreement among parental/guardian responses to measures capturing the different perceived enablers of AST are presented in Table 5. Enablers were grouped according to the three constructs noted in Table 3 (‘enjoyment of AST’, ‘supportive environment’, and ‘safe environment’). Again, the first analysis examining response on the 4-point Likert scale contained items with less reliable Kappa test statistics, as most were of ‘moderate’ or poorer (<60%) agreement. In the binary analysis, four of the eight items were of ‘moderate’ or greater agreement, with the remaining four items scoring moderate agreement in their Kappa test statistics.

### 3.3. Construct Reliability (McDonald’s Omega) and Validity (Confirmatory Factor Analyses)

Full results of the McDonald’s Omega examining the reliability of the perceived constructs are presented in Table 6, while results of the CFAs assessing the validity of the five perceived AST barrier and two perceived AST enabler constructs are presented in Table 7 (Figures of the two CFA structural models are presented in the Appendix A). Regarding the McDonald’s Omega analysis, findings suggested high levels of internal consistency regarding equipment storage (test ω = 0.916, retest ω = 0.816), road safety (test ω = 0.863, retest ω = 0.843), social safety (test ω = 0.767, retest ω = 0.818), and convenience (test ω = 0.791, retest ω = 0.822), with the skills concept scoring slightly lower (test ω = 0.610, retest ω = 0.707). Enablers were likewise evaluated across the same three concepts as the Kappa test analyses. Results of the McDonald’s Omega tests indicated that the supportive environment concept had weak internal consistency (ω = 0.430; ω = 0.465), while the feeling of safety construct had strong internal consistency (ω = 0.859; ω = 0.781). All McDonald’s Omega tests were run in SPSS using the Hayes Omega macro.

Model fit values for both CFAs generally suggest a good fit. In the five-factor perceived AST barriers CFA, model fit values were: CMIN (*p* < 0.001 (the Chi-square (CMIN) test of model fit can be sensitive to smaller sample sizes (e.g., >300) [23], thus it is possible that our smaller sample size affected this assessment metric)), CMIN/DF = 2.413, and RMSEA estimate = 0.096 (90% CI: 0.066, 0.128). With respect to the two-factor perceived AST enabler, CFA model fit values included: CMIN (*p* = 0.312), CMIN/DF = 1.191, and RMSEA estimate = 0.035 (90% CI 0.00, 0.132). Concerning discriminant validity, it has been suggested that coefficients ≥0.90 indicate high correlation between constructs and thus violate any potential discriminant validity [63]. In our CFA models, all covariances between the latent variables were <0.65 (see Appendix A for CFA structural models with latent variable coefficients), suggesting the evaluated constructs in our two CFAs exhibit sufficient discriminant validity. Standardised factor loadings are presented to illustrate the correlations between observed variables (i.e., individual items) and latent variables (i.e., AST constructs) (Ibid). Overall, the CFAs resulted in dropping 14 items from the 29 that were included in Cohen’s Kappa analyses due to their loading being <0.32, suggesting very poor correlation. Of the remaining 15 items, 13 indicated ‘very good’ or better correlations, with 10 indicating ‘excellent’ correlations. The final version of the PASTEB–P questionnaire, a tool specifically designed to assess AST-specific perceived barriers and enablers from the perspectives of parents/guardians, thus includes 15 generally highly correlated items related to five perceived barrier and two perceived enabler constructs.

## 4. Discussion

This paper presents the development and validation of the PASTEB–P questionnaire, which features multiple AST-specific constructs outlining parental/guardian perceptions of distinct socio-ecological barriers and enablers. Perceived barrier and enabler constructs were first qualitatively investigated for their face validity via literature review and cognitive interview techniques, and then subsequently statistically assessed for their reliability and construct validity. The results of this study present researchers and intervention evaluators with a validated tool containing a set of perceived parental/guardian AST barrier and enabler measures that may be used to identify and/or address specific community-, school-, or family-level perceptual phenomena in furtherance of informing programme strategy development or assessing programme effectiveness. Having developed multiple iterations of the tool, researchers are encouraged to use the final version presented in the Appendix A, whereas intervention programmers may use the broader range of measures outlined in Table 4 and Table 5 for school/community/group surveying. The ensuing discussion focuses on the methodological implications, potential research applications, and limitations of the PASTEB–P questionnaire, as well as the implications of the tool for AST-related practitioners.

### 4.1. Methodological Implications

Overall, the developed PASTEB–P questionnaire contains fifteen items related to seven distinct constructs (barriers: AST Skills, Convenience, Road Safety, Social Safety, and Equipment Storage; enablers: Supportive Environment and Safe Environment) which were validated through CFAs. Of the 15 analysed items, 13 indicated ‘very good’ factor loadings and only one relatively poor loading. It should be noted here that discussions regarding acceptable factor load cut points for newly developed items have varied considerably, e.g., [71,77], but include the range of our analysis, so we have elected to maintain the presented factor structure. More broadly, the scope of the perceptual constructs validated in the PASTEB–P represents a significant step towards developing AST-specific tools and measures that are designed to support intervention programming and research. Given that there are significant differences in how parents/guardians and children perceive barriers and enablers to AST [78], the PASTEB–P foremost addresses the need to develop reliable group-specific (i.e., parents/guardians) AST perceptual measures [22]. To ensure the suitability of the PASTEB–P as a group-specific tool, we employed a mixed-methods data collection and analysis approach that was guided by a community participatory approach featuring parents/guardians as co-creators in the initial knowledge generation processes of our study. This participatory approach situated parents/guardians as experts in the construct development phases of the qualitative work within this study, afforded them the opportunity to be actively involved in contributing to the framing questions and construct definitions, and ultimately resulted in the PASTEB–P leveraging the unique knowledge sources and assets of parents [79] to develop a relevant and authentic questionnaire. When paired with the socio-ecological underpinnings that informed the development of this tool (i.e., the original items of the questionnaire and their subsequent revisions) and often contribute to the relevant bodies of literature pertaining to various AST topics, e.g., [22,45,52], the validated constructs contained within the PASTEB–P reflect authentic parent/guardian-specific individual, interpersonal, community, and environmental AST considerations.

The collective range of items contained within the PASTEB–P includes considerations related to previously documented AST correlates such as perceptions of neighbourhood safety, pedestrian safety [28], traffic and vehicular dangers [80], street crossings, and trip companions [43], among others. These measures and constructs within the PASTEB–P reflect a set of socio-ecological constructs and appear to align well with similar existing work: a recently developed AST instrument designed to measure parental intentions examined perceived environmental barriers (e.g., sidewalk availability) and neighbourhood environment (e.g., traffic) constructs [81], and another behavioural beliefs (e.g., independence) constructs [50], while a similar previously validated Safe Routes parental survey featured a number of analogous questions regarding trip companions and perceived safety considerations [82]. Such consistency between our questionnaire and other related tools reaffirms the relevance and comprehensiveness of the PASTEB–P as an AST intervention and research tool. Moreover, with several extant objective measures (e.g., area walkability, vegetation density) often being used to predict and analyse AST topics, the scope and suitability of the PASTEB–P as a subjective perception-oriented questionnaire position it as an important AST-specific tool that can complement these objective measures. Current objective tools commonly used in AST research include the Health Economic Assessment Tool [83], the Integrated Transport and Health Impact Modelling Tool [84], and the Impacts of Cycling Tool [85]—tools that are generally more designed for population-level use and environmental assessments. The parent-specific PASTEB–P may be used alongside these tools to triangulate findings across data sources in future studies to generate more precise insights.

### 4.2. Potential Research Applications

Research focusing on children’s AST may use the validated constructs here to pursue a variety of future relevant research and intervention topics pertaining to individual, interpersonal, and environmental facets of AST. For instance, the PASTEB–P could be used to support research related to identifying and addressing subjective family travel behaviour values [24], implementing and evaluating effective family-based physical activity (i.e., AST) intervention strategies such as goal-setting and reinforcement techniques [86], or quantifying the links between different objective land use schemes (e.g., mixed use spaces, residential density) and parental/guardian perceptions of AST. Regarding the latter, the PASTEB–P could be employed to study the potential associations between parental/guardian perceptions of the local built environment and objective environmental exposures (e.g., air or noise pollution, walkability, route vegetation) to explore the potential discrepancies between perceived AST barriers/enablers and objective exposures and subsequently why these discrepancies might exist. The PASTEB–P could alternatively be used to investigate targeted intervention strategies (e.g., AST safety behaviour, engineering projects) to assess their effectiveness in mitigating barriers or reaffirming the primacy of enablers in parental/guardian decision-making processes. To this end, the barrier constructs of the PASTEB–P offer programme evaluators the potential to examine the effectiveness of AST education programmes in mitigating skill or convenience concerns among parent/guardian samples. It is also important to note that built environments can vary quite considerably with respect to their specific pedestrian features (e.g., lit/unlit street crossings, side/main street crossings, average block size) that can influence AST [87], designs that are not prominently featured in the PASTEB–P questionnaire. It is therefore advised that future research consider suitable uses for this scale, particularly those environments that feature the designs that have been validated here, such as in school intervention or health promotion contexts.

### 4.3. Implications for Practitioners

These new perceptual measures included in the PASTEB–P represent new tools for practitioners involved in community-, school-, or family-level AST programmes, such as STP, walking school buses, and school cycling programmes, to use in their programme development and evaluation processes. While other measures such as Bastem et al.’s [81] and Forsberg et al.’s [50] AST instrument to measure parental/guardian intentions offer more theoretically-oriented parental/guardian perception measures (e.g., constructs related to the Theory of Planned Behaviour), the PASTEB–P was designed with the intention of supporting AST initiatives aligned with the six Es of the Safe Routes to School National Partnership [47]. Precisely, the comparatively shorter PASTEB–P contains items and constructs related to each of the Es—engagement (e.g., enjoyment), equity (e.g., supportive environments), engineering (e.g., environment safety), encouragement (e.g., supportive environments), education (e.g., skills), and evaluation (the tool itself)—and thus offers programme designers and evaluators a coherent set of reliable and suitable measures to evaluate the effectiveness of their interventions across these recognised aims and criteria. Additionally, recognising that parental/guardian decision-making processes regarding AST tend to be characterised by convenience and the relative ease of commute considerations [88], the PASTEB–P’s component items present a reliable, valid, and efficient means for programmers to assess parental/guardian perceptions regarding a potential intervention’s reach, precision, efficacy, or effectiveness. Lastly, as recent work has indicated that the quality of AST practises and the extent to which they are voiced and embedded within a local community and school culture are important factors to consider in promoting AST [89], the PASTEB–P’s environmental measures can be used to provide further insights into these community culture dynamics to inform local programme design. Overall, owing to the social-ecological framing of the tool, the PASTEB–P’s various measures can be used to inform the development and/or selection of AST support strategies for schools or local school communities. For instance, the expanded array of measures outlined in Table 4 and Table 5 could be used to survey target groups to explore the relative prominence or influence of local AST issues such as family behaviours and beliefs [54] (e.g., convenience measures), perceptions of neighbourhood safety [57] (e.g., social safety measures), or infrastructural quality [59] (e.g., supportive environment measures).

### 4.4. Limitations

There are a few limitations to the PASTEB–P questionnaire. First, the timeline for the Test-Retest Study was 5–7 days between surveys, which reflects a relatively short timeframe. Consequently, this timeline may not accurately reflect the variance in commuting schedules that exist in a typical community (e.g., in real-world settings, some families exclusively passively commute and are unlikely to remember specific AST barriers). This particular reporting dynamic in the Test-Retest Study may have introduced some desirability bias that negatively affected the true reliability of the tested measures, as otherwise unconcerned participants may have been biassed by the short distance in time between the surveys. Likewise, this study protocol may have subjected the final results to some level of testing response bias wherein previous familiarity with the survey may have influenced participant response patterns during the follow-up (i.e., retest) survey (i.e., within-subject threat to internal validity). Relatedly, our data collection for both studies involved in the scale development was completed during the pandemic, which necessitated participants having to answer based on best approximations of behaviours when their children were in school. As a result of this dynamic, the responses recorded in the two component studies may be subject to a relatively greater risk of recall bias (as opposed to studies not conducted during the pandemic). While our study focused on parental perceptions as they related to children ages 9–14, a group that is comparatively small relative to other scales that have been designed for similar topics [50], e.g., ages 5–18, we also acknowledge that there is other data that highlights that there can be important differences among those in this cohort [90,91]. We therefore caution against the use of the scale for groups exclusively at the tails of the sample (i.e., samples exclusively composed of 9- or 13-/14-year-olds). Last, the recruited samples for both studies were relatively well-educated, born in Canada, and featured female parent samples, which limits the generalizability of the paper’s findings with respect to males, gender-diverse parents/guardians, and less educated and recent immigrant families.

## 5. Conclusions

In this paper, a set of parental/guardian-specific perceived barriers and enablers of AST constructs were constructed and validated through a mixed approach featuring a qualitative thematic analysis, followed by a series of statistical methods including Cohen’s Kappa, McDonald’s Omega, and confirmatory factor analysis (CFA). Ultimately, fifteen items comprising seven distinct constructs (barriers: AST Skills, Convenience, Road Safety, Social Safety, and Equipment Storage; enablers: Supportive Environment and Safe Environment) were included and validated in the final CFA, forming the PASTEB–P questionnaire, which is suitable for use in the context of AST research. As this represents one of the first scales to validate the AST-specific considerations of parents/guardians, future AST scholarship may use this questionnaire to explore parental/guardian decision-making processes or to compare the influence of subjective parental/guardian perceptions with objective environmental factors regarding AST behaviours. In the same vein as this manuscript, future methodological work should seek to validate child-perceived AST barriers and enablers, for instance by validating some of the AST constructs explored here by way of other methods (e.g., objectively measured volumes or traffic density for “busy roads”), with more varied groups of parents/guardians (i.e., more males/fathers included), or in reference to other interpretations or forms of specific constructs (e.g., “safety” from environmental pollutants).

## Figures and Tables

**Table 1 ijerph-20-05874-t001:** Description of Child-Level Travel Behaviour, n (%).

Variable	Validation Pilot Study	Test-Retest Pilot Study
Sample Size, N	136	152
Children living within walking distance ^a^	70 (51.5)	72 (47.0)
**Predominant Mode of Travel to School**		
Active School Travel	48 (35.3)	64 (41.8)
Car	25 (18.4)	39 (25.5)
School Bus	48 (35.3)	36 (23.5)
**Predominant Mode of Travel Home from School**		
Active School Travel	58 (42.6)	72 (47.1)
Car	22 (16.2)	47 (30.7)
School Bus	48 (35.3)	23 (15.0)

Note: ^a^ walking distance was operationalized as <1.6 km, which is the regional bus eligibility limit.

**Table 2 ijerph-20-05874-t002:** Sample characteristics of the participants in the (a) Validation Pilot Study and (b) Test-Retest Pilot Study.

Variable	Validation Pilot Study	Test-Retest Pilot Study
**Child Characteristics, N**	136	152
Age in years, mean (SD)	9.9 (2.3)	10.1 (1.4)
Grade, mean (SD)	4.6 (2.2)	5.0 (1.4)
**Gender, n (%)**		
Boy	63 (46.3)	78 (51.0)
Girl	73 (53.7)	73 (47.7)
Non-Binary	0 (0.0)	1 (0.7)
**Race, n (%)**		
Black	3 (2.2)	0 (0.0)
Caucasian	103 (75.7)	107 (69.9)
East and Southeast Asian	7 (5.1)	9 (5.2)
Indigenous	2 (1.5)	5 (3.3)
Latinx	3 (2.2)	3 (2.0)
South Asian	3 (2.2)	8 (5.2)
West Asian	2 (1.5)	2 (1.3)
Mixed Race	13 (9.6)	18 (11.8)
**Immigration Status, n (%)**		
Born in Canada	124 (91.2)	143 (93.5)
Born Outside of Canada	12 (8.8)	10 (6.5)
**Parent/Guardian Characteristics, N**	80	86
**Gender, n (%)**		
Man	9 (11.3)	10 (11.6)
Woman	71 (88.8)	75 (87.2)
Non-Binary	0 (0.0)	1 (1.2)
**Highest educational attainment, n (%)**		
Did not graduate high school	0 (0.0)	1 (1.2)
High School	4 (5.0)	2 (2.3)
Apprenticeship/Trade Certificate	2 (2.5)	1 (1.2)
Post Secondary	44 (55.1)	60 (69.7)
Postgraduate Programme	30 (37.5)	21 (24.4)
Children live in two homes, n (%)	6 (7.5)	11 (12.8)
Children live in homes led by a lone parent/guardian, n (%)	9 (12.2)	14 (16.6)

Note: The child characteristics are based on a sample of all children in Junior Kindergarten to grade 8 from the family that attended the elementary school identified by the parent.

**Table 3 ijerph-20-05874-t003:** Survey questions used to assess parents’/guardians’ perceptions of barriers and enablers of active school travel used for the validation pilot study.

**Barriers to Active School Travel**	
**Questions for the Validation Pilot Study**	**Questions for the Test-Retest Pilot Study**
**Skills:**	
My child does not have the skills to bike.	My child does not have the skills to bike.
My child does not have the skills to walk.	My child does not have the skills to walk.
My child does not have the skills to roll.	My child does not have the skills to roll.
My child is too young to walk/bike/roll.	My child is too young to walk/bike/roll.
**Convenience:**	
It is too far for my child to walk.	It is too far for my child to walk/bike/roll.
We do not have enough time in the morning.	We do not have enough time in the morning.
We do not have enough time in the afternoon.	We do not have enough time in the afternoon.
The parent/guardian schedule does not allow for it.	The parent/guardian schedule does not allow for it.
My child does not like to walk.	REMOVED
My child has too much stuff to carry.	REMOVED
It is too cold in the winter.	REMOVED
It is too hot in the spring/fall.	REMOVED
My child gets too hot/sweaty.	REMOVED
**Road Safety:**	
There are not enough sidewalks along the route.	Not enough sidewalks along the route to school.
There are not enough crosswalks along the route.	Not enough crosswalks along the route to school.
There is nowhere for my child to safely leave a bike at school.	Nowhere for my child to safely leave their bike at school.
There is nowhere for my child to safely leave their scooter, skateboard, or rollerblades.	Nowhere for my child to safely leave their scooter at school.
The sidewalks along the route are not maintained in the winter.	The sidewalks are not cleared along the route to school.
Drivers drive too fast.	Drivers drive too fast.
Too much traffic on the route.	Too much traffic on the route.
A child must cross busy roads.	A child must cross busy roads.
**Social Safety:**	
Unsafe because of crime in our neighbourhood.	Unsafe because of crime in our neighbourhood.
Unsafe because of strangers in the neighbourhood.	Unsafe because of strangers in the neighbourhood.
Might get bullied/teased.	Might get bullied/teased.
No adults to walk with.	No adults/high school students to walk with.
No children to walk with.	No peers (e.g., friends and siblings at the school) to walk with.
**Daylight:**	
There is not enough daylight in the morning.	REMOVED
There is not enough daylight in the afternoon.	REMOVED
**Equipment Storage:**	
ADDED	Nowhere for my child to safely leave their bike at school.
ADDED	Nowhere for my child to safely leave their scooter at school.
**Enablers for Active School Travel:**	
**Questions for theValidation Pilot Study**	**Questions for the Test-Retest Pilot Study**
**Enjoyment of AST:**	
I chose to live in my area so that my child could walk/bike/roll.	I chose to live in my area so my child could walk/bike/roll to and from school.
My child enjoys walking/biking/rolling for fun outside of school hours.	REMOVED
My child enjoys [walking/biking/rolling to school] OR [walking to/from the bus stop].	REMOVED
**Supportive Environment:**	
There are many interesting things to look at along the way.	REMOVED
There are lots of trees along the route.	There are lots of trees along the route between home and school.
There are enough sidewalks along the route.	There are enough sidewalks along the route between home and school.
There are walking trails along the route.	There are walking paths and cut-throughs to shorten the route between home and school.
**Safe Environment:**	
My neighbourhood is safe enough for children to walk to/from school alone.	My neighbourhood is safe enough for children to walk/bike/roll to and from school alone.
My neighbourhood is safe enough for children to walk to/from school with friends.	My neighbourhood is safe enough for children to walk/bike/roll to and from school with friends.
Pedestrians/cyclists can be seen by people in their homes along the way.	My children are visible to my neighbours when walking, biking, and rolling along their route to and from school.
ADDED	There are crossing guards to help my child cross the street on the way to school.

**Table 4 ijerph-20-05874-t004:** Results of the Cohen’s Kappa test of the barriers to active school travel.

Barriers to Active School Travel		Responses				4-Point Likert Scale		Binary Scale	
		Strongly Agree	Agree	Disagree	Strongly Disagree	% Agree	Kappa [95% CI]	% Agree	Kappa [95% CI]
**Skills:**									
My child does not have the skills to bike.	Test	9 (9.3%)	12 (12.4%)	24 (24.7%)	52 (53.6%)	58.8%	0.351 [0.220–0.483]	92.8%	0.784 [0.585–0.983]
	Retest	6 (6.2%)	14 (14.4%)	29 (29.9%)	48 (49.5%)				
My child does not have the skills to walk.	Test	0 (0%)	2 (2.1%)	13 (13.4%)	82 (84.5%)	72.2%	0.088 [−0.084–0.260]	97.9%	0.656 [0.457–0.855]
	Retest	1 (1%)	3 (3.1%)	16 (16.5%)	77 (79.4%)				
My child does not have the skills to roll.	Test	14 (14.4%)	11 (11.3%)	23 (23.7%)	49 (50.5%)	58.8%	0.370 [0.241–0.498]	87.6%	0.647 [0.448–0.846]
	Retest	8 (8.2%)	11 (11.3%)	32 (33%)	46 (47.4%)				
My child is too young to walk/bike/roll.	Test	8 (8.2%)	23 (23.7%)	38 (39.2%)	28 (28.9%)	58.8%	0.419 [0.294–0.543]	72.2%	0.393 [0.194–0.592]
	Retest	8 (8.2%)	30 (30.9%)	30 (30.9%)	29 (29.9%)				
**Convenience:**									
It is too far for my child to walk/bike/roll.	Test	9 (9.5%)	22 (23.2%)	20 (21.1%)	44 (46.3%)	63.2%	0.447 [0.321–0.573]	86.3%	0.660 [0.459–0.861]
	Retest	12 (12.6%)	10 (10.5%)	24 (25.3%)	49 (51.6%)				
We do not have enough time in the morning.	Test	8 (8.2%)	25 (25.8%)	26 (26.8%)	38 (39.2%)	56.7%	0.387 [0.265–0.510]	81.4%	0.587 [0.388–0.786]
	Retest	12 (12.4%)	21 (21.6%)	25 (25.8%)	39 (40.2%)				
We do not have enough time in the afternoon.	Test	8 (8.2%)	11 (11.3%)	30 (30.9%)	48 (49.5%)	66.0%	0.471 [0.337–0.605]	84.5%	0.499 [0.300–0.698]
	Retest	9 (9.3%)	9 (9.3%)	34 (35.1%)	45 (46.4%)				
The parent/Guardian schedule does not allow for it.	Test	8 (8.2%)	22 (22.7%)	24 (24.7%)	43 (44.3%)	56.7%	0.374 [0.249–0.499]	74.2%	0.402 [0.203–0.601]
	Retest	9 (9.3%)	22 (22.7%)	26 (26.8%)	40 (41.2%)				
**Road Safety:**									
Not enough sidewalks along the route to school.	Test	10 (10.3%)	24 (24.7%)	30 (30.9%)	33 (34%)	67.0%	0.532 [0.409–0.656]	84.5%	0.635 [0.436–0.834]
	Retest	9 (9.3%)	16 (16.5%)	40 (41.2%)	32 (33%)				
Not enough crosswalks along the route to school.	Test	14 (14.4%)	22 (22.7%)	41 (42.3%)	20 (20.6%)	55.7%	0.362 [0.241–0.484]	76.3%	0.483 [0.284–0.682]
	Retest	11 (11.3%)	22 (22.7%)	45 (46.4%)	19 (19.6%)				
The sidewalks are not cleared along the route to school.	Test	11 (11.8%)	38 (40.9%)	29 (31.2%)	15 (16.1%)	71.1%	0.555 [0.420–0.689]	91.8%	0.738 [0.539–0.937]
	Retest	14 (15.1%)	32 (34.4%)	36 (38.7%)	11 (11.8%)				
Drivers drive too fast.	Test	35 (36.1%)	42 (43.3%)	11 (11.3%)	9 (9.3%)	59.8%	0.412 [0.288–0.536]	80.4%	0.528 [0.329–0.727]
	Retest	31 (32%)	48 (49.5%)	13 (13.4%)	5 (5.2%)				
Too much traffic on the route.	Test	27 (27.8%)	46 (47.4%)	12 (12.4%)	12 (12.4%)	64.9%	0.506 [0.385–0.626]	86.6%	0.699 [0.500–0.898]
	Retest	22 (22.7%)	42 (43.3%)	25 (25.8%)	8 (8.2%)				
A child must cross busy roads.	Test	26 (26.8%)	39 (40.2%)	18 (18.6%)	14 (14.4%)	57.0%	0.384 [0.258–0.511]	73.1%	0.462 [0.259–0.665]
	Retest	24 (24.7%)	40 (41.2%)	21 (21.6%)	12 (12.4%)				
**Social Safety:**									
Unsafe because of crime in our neighbourhood.	Test	0 (0%)	3 (3.1%)	40 (41.7%)	53 (55.2%)	76.0%	0.559 [0.383–0.735]	96.9%	0.650 [0.450–0.850]
	Retest	3 (3.1%)	3 (3.1%)	47 (49%)	43 (44.8%)				
Unsafe because of strangers in the neighbourhood.	Test	0 (0%)	27 (28.4%)	35 (36.8%)	33 (34.7%)	58.9%	0.360 [0.214–0.505]	78.9%	0.389 [0.188–0.590]
	Retest	0 (0%)	15 (15.8%)	50 (52.6%)	30 (31.6%)				
Might get bullied/teased.	Test	1 (1%)	15 (15.5%)	42 (43.3%)	39 (40.2%)	61.9%	0.360 [0.214–0.505]	89.7%	0.583 [0.384–0.782]
	Retest	0 (0%)	12 (12.4%)	54 (55.7%)	31 (32%)				
No adults/high school students to walk with.	Test	5 (5.2%)	21 (21.6%)	39 (40.2%)	32 (33%)	59.8%	0.370 [0.220–0.520]	84.5%	0.601 [0.402–0.800]
	Retest	7 (7.2%)	18 (18.6%)	42 (43.3%)	30 (30.9%)				
No peers (e.g., friends and siblings at the school) to walk with.	Test	10 (10.3%)	26 (26.8%)	30 (30.9%)	31 (32%)	64.9%	0.515 [0.393–0.634]	91.8%	0.819 [0.62–1.018]
	Retest	13 (13.4%)	19 (19.6%)	36 (37.1%)	29 (29.9%)				
**Equipment Storage:**									
Nowhere for my child to safely leave their bike at school.	Test	9 (11.3%)	14 (17.5%)	38 (47.5%)	19 (23.8%)	68.8%	0.523 [0.383–0.663]	85.0%	0.613 [0.393–0.832]
	Retest	5 (6.3%)	14 (17.5%)	42 (52.5%)	19 (23.8%)				
Nowhere for my child to safely leave their scooter at school.	Test	11 (13.9%)	29 (36.7%)	26 (32.9%)	13 (16.5%)	63.3%	0.480 [0.343–0.616]	82.3%	0.646 [0.425–0.866]
	Retest	9 (11.4%)	29 (36.7%)	28 (35.4%)	13 (16.5%)				

**Table 5 ijerph-20-05874-t005:** Results of the Cohen’s Kappa test of the enablers of active school travel.

Enablers of Active School Travel:		Responses				4-Point Likert Scale		Binary Scale	
		Strongly Agree	Agree	Disagree	Strongly Disagree	% Agree	Kappa [95% CI]	% Agree	Kappa [95% CI]
**Enjoyment of AST:**									
I chose to live in my area so my child could walk/bike/roll to and from school.	Test	39 (41.5%)	22 (23.4%)	27 (28.7%)	6 (6.4%)	80.9%	0.718 [0.588–0.849]	91.5%	0.807 [0.605–1.010]
	Retest	43 (45.7%)	22 (23.4%)	23 (24.5%)	6 (6.4%)				
**Supportive Environment:**									
There are lots of trees along the route between home and school.	Test	21 (22.3%)	44 (46.8%)	25 (26.6%)	4 (4.3%)	62.8%	0.455 [0.325–0.586]	83.0%	0.621 [0.419–0.823]
	Retest	23 (24.5%)	36 (38.3%)	27 (28.7%)	8 (8.5%)				
There are enough sidewalks along the route between home and school.	Test	35 (37.2%)	45 (47.9%)	9 (9.6%)	5 (5.3%)	69.1%	0.515 [0.375–0.656]	83.0%	0.426 [0.223–0.628]
	Retest	30 (31.9%)	44 (46.8%)	16 (17%)	4 (4.3%)				
There are walking paths and cut-throughs to shorten the route between home and school.	Test	26 (27.7%)	27 (28.7%)	23 (24.5%)	18 (19.1%)	63.8%	0.511 [0.392–0.629]	87.2%	0.743 [0.541–0.945]
	Retest	21 (22.3%)	28 (29.8%)	32 (34%)	13 (13.8%)				
**Safe Environment:**									
My neighbourhood is safe enough for children to walk/bike/roll to and from school alone.	Test	23 (24.5%)	53 (56.4%)	14 (14.9%)	4 (4.3%)	71.3%	0.505 [0.361–0.649]	84.0%	0.449 [0.247–0.651]
	Retest	23 (24.5%)	56 (59.6%)	15 (16%)	0 (0%)				
My neighbourhood is safe enough for children to walk/bike/roll to and from school with friends.	Test	33 (37.5%)	49 (55.7%)	2 (2.3%)	4 (4.5%)	65.9%	0.400 [0.235–0.565]	92.0%	0.544 [0.335–0.753]
	Retest	30 (34.1%)	47 (53.4%)	11 (12.5%)	0 (0%)				
My children are visible to my neighbours when walking, biking, and rolling along their route to and from school.	Test	24 (27.3%)	52 (59.1%)	5 (5.7%)	7 (8%)	68.2%	0.433 [0.281–0.584]	89.8%	0.550 [0.341–0.759]
	Retest	24 (27.3%)	53 (60.2%)	8 (9.1%)	3 (3.4%)				
There are crossing guards to help my child cross the street on the way to school.	Test	11 (12.5%)	29 (33%)	14 (15.9%)	34 (38.6%)	70.5%	0.579 [0.449–0.710]	93.2%	0.863 [0.654–1.071]
	Retest	7 (8%)	33 (37.5%)	19 (21.6%)	29 (33%)				

**Table 6 ijerph-20-05874-t006:** Results of McDonald’s Omega analysis of the barriers/enablers to active school travel.

	McDonald’s Omega (ω)	
	Test	Retest
**Constructs of Barriers**		
Skills	0.610	0.707
Convenience	0.791	0.822
Road Safety	0.863	0.843
Social Safety	0.767	0.818
Equipment Storage	0.916	0.816
**Constructs of Enablers**		
Supportive Environments	0.430	0.465
Feeling of Safety	0.859	0.781

**Table 7 ijerph-20-05874-t007:** Confirmatory Factor Analysis of the PASTEB–P questionnaire (15 Items).

Construct	Item	Item Question	Standardized Factor Loading
**Barriers**			
AST Skills	Bike Skills	My child does not have the skills to bike.	0.71
	Rolling Skills	My child does not have the skills to roll.	0.71
Convenience	Too Busy (Morn.)	We do not have enough time in the morning.	0.99
	Too Busy (Aft.)	We do not have enough time in the afternoon.	0.66
Road Safety	Traffic	Too much traffic on the route.	0.65
	Crossing Roads	A child must cross busy roads.	0.73
Social Safety	No Adults	No adults/high school students to walk with.	0.75
	No Peers	No peers (e.g., friends and siblings at the school) to walk with.	0.99
Equipment	No Bike Storage	Nowhere for my child to safely leave their bike at school.	0.59
	No Scooter Storage	Nowhere for my child to safely leave their scooter at school	0.86
**Enablers**			
Supportive Environment	Enough Sidewalks	There are enough sidewalks along the route between home and school.	0.37
	Shortcuts	There are walking paths and cut-throughs to shorten the route between home and school.	0.91
Safe Environment	Safe Alone	My neighbourhood is safe enough for children to walk/bike/roll to and from school alone.	0.82
	Safe with Friends	My neighbourhood is safe enough for children to walk/bike/roll to and from school with friends.	0.81
	Child Visible	My children are visible to my neighbours when walking, biking, and rolling along their route to and from school.	0.65

## Data Availability

For participant privacy reasons associated with our data storage and management plans we are unable to make available our data.

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
