# Peer review of "Validating the Perceived Active School Travel Enablers and Barriers–Parent (PASTEB–P) Questionnaire to Support Intervention Programming and Research"

_ijerph, 2023, doi:10.3390/ijerph20105874_

Round 1

Reviewer 1 Report (New Reviewer)

The development of validated questionnaires to measure parental influences on active school travel is very important. I applaud the effort but I have a number of comments and questions:

1.     I would contest the use of the phrase “perhaps most consequential”. We do not have evidence for this, especially for the large range of 9-14 year-olds. Consequential is fine because we do have evidence. Just how parental attitudes interact with local environment remains mysterious, so it is quite conceivable that we exaggerate parental influence unless we account for moderators.

2.     I cannot find the long form of the acronym STP. Perhaps I missed it after two readings.

3.     I agree with the limitation note that the test and retest were quite close in time, but also a response to this kind of question is highly likely to be remembered. But ok. I think more important is the test period, May 2021 to March 2022. I am unsure of the conditions in the test environment, but to my knowledge, children were not in school or were beginning to return to school. The parents/guardians were responding hypothetically or with regard to their memory of the previous year/s when their children were younger. I also wonder whether the use of the generic “safe” might also include safety from infection. 

4.     The children of the respondents were between 9 and 14 years of age. My own studies show enormous differences in mobility between 9 and 12. For the purposes of a questionnaire, the concerns of parents might be quite different as the children pass into adolescence. I would guess concerns about road safety will decline while new concerns about the activities of their charges emerge.

5.     Following 3. above, I wonder about the understanding that parents/guardians have about the concept of safety. The range of possible issues seems enormous and might also be gender-specific. Agreement on the measure does not include agreement on the concept.

6.     I was surprised by the measure “enough sidewalks”. Maybe this would apply in southern Ontario, but certainly not in many environments where sidewalks are included when traffic volumes or speeds are high. A pedestrian street doesn’t have sidewalks and most gated communities don’t have them either. There is no measure for the number of street crossings and lighted/unlighted ones. Of course we could expand the list of possible measures to be inclusive of many environments, but I just mention this because the proposed measures are quite specific to particular environments.

7.     The following is just a suggestion for further development in a subsequent stage of work: As with many of these qualitative concepts, we do not know how they translate, or if they do, into objective measures. “Too much traffic” or “busy roads” could also be validated with examples and specific, objectively measured volumes or traffic density or something other. 

8.     Finally, the parent/guardian group is almost entirely female. I know this was noted in the Limitations, but it seems quite a crucial matter. Perhaps the range of response might vary by sex, and this would not specifically be a problem in the present work of research instrument development. Males and females may have remarkably different ideas about what is important for the safety and health of their children, and those concerns will vary by the sexes of the parent/child pairs.

I see this questionnaire as a good initial effort with limited applicability. The absence of these instruments generally in the field makes me lean slightly in favour of publication; otherwise, I would want to see more validation, an elaboration on the concepts and a more comprehensive test instrument.

Author Response

Reviewer 1

The development of validated questionnaires to measure parental influences on active school travel is very important. I applaud the effort but I have a number of comments and questions:

  1. I would contest the use of the phrase “perhaps most consequential”. We do not have evidence for this, especially for the large range of 9-14 year-olds. Consequential is fine because we do have evidence. Just how parental attitudes interact with local environment remains mysterious, so it is quite conceivable that we exaggerate parental influence unless we account for moderators.

Response: Thank you for this note. We have now changed this phrasing to “Of particular note”.

  1. I cannot find the long form of the acronym STP. Perhaps I missed it after two readings.

Response: This acronym has been outlined in the line of the introduction section.

  1. I agree with the limitation note that the test and retest were quite close in time, but also a response to this kind of question is highly likely to be remembered. But ok. I think more important is the test period, May 2021 to March 2022. I am unsure of the conditions in the test environment, but to my knowledge, children were not in school or were beginning to return to school. The parents/guardians were responding hypothetically or with regard to their memory of the previous year/s when their children were younger. I also wonder whether the use of the generic “safe” might also include safety from infection. 

Response: Thank you for bringing these details to our attention. First, we have now added the following text to the limitations in response to this point in the above feedback: “Relatedly, our data collection for both of the studies involved in the scale development was completed during the pandemic which necessitated participants having to answer based on best approximations of behaviours when their children were in school. As a result of this dynamic, the responses recorded in the two component studies may be subject to a relatively greater risk of recall bias (as opposed to studies not conducted during the pandemic).” And second, regarding the word safe, cognitive interviews were used to ensure consistent interpretation of words across participants. In this case, the word safe was deemed most appropriate from the analysis. However, to incorporate this feedback, we have added a note on this point in the concluding paragraph of the revised manuscript: “In the same vein as this manuscript, future methodological work should seek to validate child perceived AST barriers and enablers, for instance via validating some of the AST constructs explored here by way of other methods (e.g., objectively measured volumes or traffic density for “busy roads”), with more varied groups of parents/guardians (i.e., more males/fathers included), or in reference to other interpretations or forms of specific constructs (e.g., “safety” from environmental pollutants).” (emphasis only added here)

  1. The children of the respondents were between 9 and 14 years of age. My own studies show enormous differences in mobility between 9 and 12. For the purposes of a questionnaire, the concerns of parents might be quite different as the children pass into adolescence. I would guess concerns about road safety will decline while new concerns about the activities of their charges emerge.

Response: Thank you for this detailed note. In response to this feedback we have now discussed this issue in our revised limitations section with the following text: “While our study focused on parental perceptions as they related to children ages 9-14, a group which is comparatively small relative to other scales that have been designed for similar topics [50, e.g., ages 5-18], we also acknowledge that there is other data which highlights there can be important differences among those in this cohort [93,94]. We therefore caution against the use of the scale for groups exclusively at the tails of the sample (i.e., samples exclusively composed of 9- or 13/14-year-olds).”

  1. Following 3. above, I wonder about the understanding that parents/guardians have about the concept of safety. The range of possible issues seems enormous and might also be gender-specific. Agreement on the measure does not include agreement on the concept.

Response: Regarding the feedback pertaining to the word safe, we refer to the response above where this issue was previously discussed. Similarly, concerning the gender aspect of this comment, the same addition to the conclusion also features a suggestion for more varied gender-specific research going forward. While we certainly appreciate the specificity of this feedback, to fully incorporate these revisions into the revised manuscript, however, this material is beyond the scope of this paper. Consequently, we have suggested these points be picked up in future research opportunities.

  1. I was surprised by the measure “enough sidewalks”. Maybe this would apply in southern Ontario, but certainly not in many environments where sidewalks are included when traffic volumes or speeds are high. A pedestrian street doesn’t have sidewalks and most gated communities don’t have them either. There is no measure for the number of street crossings and lighted/unlighted ones. Of course we could expand the list of possible measures to be inclusive of many environments, but I just mention this because the proposed measures are quite specific to particular environments.

Response: Thank you for bringing this point to our attention. In the revised manuscript we have added a note on this point to our potential research applications sections (i.e., section 4.2) via the following text: “It is also important to note that built environments can vary quite considerably with respect to their specific pedestrian features (e.g., lit/unlit street crossings, side/main street crossings, average block size) that can influence AST [90], designs which aren’t prominently featured in the PASTEB–P questionnaire. It is therefore advised that future research considers suitable uses for this scale, particularly those environments which feature the designs that have been validated here, such as in school intervention or health promotion contexts.”

  1. The following is just a suggestion for further development in a subsequent stage of work: As with many of these qualitative concepts, we do not know how they translate, or if they do, into objective measures. “Too much traffic” or “busy roads” could also be validated with examples and specific, objectively measured volumes or traffic density or something other. 

Response: Thank you for this suggestion. We have now added it as a suggestion for future research opportunities at the end of our conclusion in the following text: “In the same vein as this manuscript, future methodological work should seek to validate child perceived AST barriers and enablers, for instance via validating some of the AST constructs explored here by way of other methods (e.g., objectively measured volumes or traffic density for “busy roads”), with more varied groups of parents/guardians (i.e., more males/fathers included), or in reference to other interpretations or forms of specific constructs (e.g., “safety” from environmental pollutants).” (emphasis only added here)

  1. Finally, the parent/guardian group is almost entirely female. I know this was noted in the Limitations, but it seems quite a crucial matter. Perhaps the range of response might vary by sex, and this would not specifically be a problem in the present work of research instrument development. Males and females may have remarkably different ideas about what is important for the safety and health of their children, and those concerns will vary by the sexes of the parent/child pairs. I see this questionnaire as a good initial effort with limited applicability. The absence of these instruments generally in the field makes me lean slightly in favour of publication; otherwise, I would want to see more validation, an elaboration on the concepts and a more comprehensive test instrument.

Response: Thanks for this note. We have also now suggested that this point regarding the gender imbalance of our sample as a limitation of this work as well as something to pursue in future research at the end of our conclusion.

Reviewer 2 Report (New Reviewer)

The manuscript is an interesting study. It is especially important for the healthy development of children and adolescents. The overall design of this study is sound and logical, and it has important practical significance and value in the development of current life health. However, there are some problems with the manuscript, which should be revised and improved in detail before accepting it for publication.

First, the preface section of the article should further explain the significance and value of the study. Valuable references need to be added as support for this study.

Secondly, the research method section does not introduce the process of subject extraction in sufficient detail. The description of the data statistical methods section of the manuscript is not detailed enough, and the data processing methods should be further detailed so that the readers can understand the data analysis process of the manuscript more clearly and increase the readability of the article.

Third, the discussion section of the manuscript is not deep enough, and the corresponding references should be added to analyze the specific reasons that led to the results of this study in depth again.

Author Response

Reviewer 2

The manuscript is an interesting study. It is especially important for the healthy development of children and adolescents. The overall design of this study is sound and logical, and it has important practical significance and value in the development of current life health. However, there are some problems with the manuscript, which should be revised and improved in detail before accepting it for publication.

First, the preface section of the article should further explain the significance and value of the study. Valuable references need to be added as support for this study.

Response: Thank you for this feedback. As a part of revisions undertaken in response to other feedback, we have added a new section to the introduction (see section 1.2 – Theoretical Background) which features several additional references to support the general contribution of this manuscript, as well as several new references throughout the methods (e.g., McDonald’s Omega analysis) and discussion (e.g., Research Applications, Limitations) sections.

Secondly, the research method section does not introduce the process of subject extraction in sufficient detail. The description of the data statistical methods section of the manuscript is not detailed enough, and the data processing methods should be further detailed so that the readers can understand the data analysis process of the manuscript more clearly and increase the readability of the article.

Response: Thanks for this note. Again, in response to other feedback, we have made several revisions to the methods section to more clearly outline these aspects of the study. Specifically, as highlighted by other reviewers, we have both changed the reliability analysis (from Cronbach’s Alpha to McDonald’s Omega) and added in an expanded description of this analysis (please see section 2.3.3.).

Third, the discussion section of the manuscript is not deep enough, and the corresponding references should be added to analyze the specific reasons that led to the results of this study in depth again.

Response: In addition to the many references that have been added to the revised manuscript in the introduction section, as well as the methods section, we have also added several new references to the discussion. Notably, as a result of feedback from the other reviewers, we have added references to the limitations of study (section 4.4), the research applications section (section 4.2), and the implications for practitioners section (4.3).

Reviewer 3 Report (New Reviewer)

This article is well written and add valuable information to existing knowledge in this area but there are some improvements that the authors need to provide before it is ready to be published.  

Suggestions of improvements:

Title: Please be consistent or provide an explanation to why you use enablers in the title but facilitators in the text. Moreover, the aim of this study does not seem coherent with the title, please look over this.  

Abstract: It is not possible to find the results of the Cronbach Alpha analysis in the manuscript? Please, overlook this.

Introduction:

Be consistent of the use of concepts used in the text, for example why do the authors use detriment and not barrier? Sometimes you use social barriers, sometimes socio-ecological facilitators and barriers. The same goes for enablers and facilitators.

In the rational of doing this study you argue that earlier studies of parental perceptions of AST focused primarily on parental intentions and behavioural beliefs, why is this a limit?  In the argumentation of doing this study you also mention the lack of social barriers. In your instrument it seems like you use the term social safety, are these different and then how does instrument contribute to the lack of social barriers? Please provide an explanation for this.

In the aim you use social and physical environment items and in the method section facilitators and barriers? Unify or provide and explanation why

Aim:

The aim is already presented in last paragraph? It would be clearer to the reader if the aims were at one place only, using the same adjectives.

Please provide a argument for not using theory in the development phase of the instrument. As this is otherwise what is recommended and really important. If you are not able to provide this explanation, please make this evident in the limitations of the study.

You state in the gap that social barriers are missing in current instruments. But here you only mention barriers not social barriers. In the arguments for this study you claim that other studies do not include social barriers, they do in fact include barriers. In one paper these items are defined as impeding factors and not barriers. Please make this clear.

Method

Data collection: There no such thing as a neutral answer option, because a score on a 3 will be more than 1 right? I dont think this sentence benefits your paper. However, it is an excellent decision to use 4- point likert scale. Please revise this.

Table 3: This is a really nice table, making it possible for the reader to follow the development phase, which is hardly ever seen.

Data analysis: In this section the Cronbach Alpha that is stated to be used in the abstract is missing. It would for sure strengthen the paper to add an analysis of the internal consistency. Although we would recommend the authors to use the McDonald Omega for likert-scales. Please revise this.

Confirmatory factor analysis: It is not clear why you use the CFA. The CFA can be used if 1, if there is a clear theory of how the items are supposed to act, because you want to confirm that model or 2, as a second step in the validation process of an instrument that has previously been explored with exploratory factor analysis (EFA)? We think it would strengthen your paper to add sentences supporting this choice. Please provide an explanation for the use of this analysis in the development phase or revise the analysis.

If the CFA is done separate for facilitators and barriers we are not sure if it is possible to claim that one instrument has been developed. Rather this is two separate instruments. The authors may need to elaborate on this issue.

Please motivate choosing only two goodness of fit indicies. You use AMOS and this will provide more than this goodness of fit indices.  Why not report them or decide to a reference the choice made.

Please add which version of AMOS

Results

Reliability and Consistency: It is hard as reader to understand why the authors have chosen to dichotomize these items. If the items are going to be used this way, please add this in your paper. If the items are not going to be used this way this is not a part that strengthens your paper, as you risk loosing information in data when dichotomizing and that may be the reason for Kappa values being higher. So that does not in fact improve the stability if your instrument. 

Appendix A: It would benefit your paper to redo these figures. These comes directly from the Software, still with names of items used in the software?

Author Response

Reviewer 3

This article is well written and add valuable information to existing knowledge in this area but there are some improvements that the authors need to provide before it is ready to be published.  

Suggestions of improvements:

Title: Please be consistent or provide an explanation to why you use enablers in the title but facilitators in the text. Moreover, the aim of this study does not seem coherent with the title, please look over this.  

Response: Thank you for pointing out this inconsistency. We have gone through the revised manuscript and changed all instances of the word facilitator with enabler. Additionally, we have now revised our aim to be simpler in section 1.3 with the following text: “Due to the uncertainties pertaining to the validity of AST perception constructs and measures, this study aims to assess the construct validity and internal consistency of AST-specific measures reflecting barriers and enablers to participation from the perspective of parents.”

Abstract: It is not possible to find the results of the Cronbach Alpha analysis in the manuscript? Please, overlook this.

Response: Thanks for highlighting this oversight. In line with other feedback, we have now removed the Cronbach’s Alpha, and in its place we have added a McDonald’s Omega analysis.

Introduction: Be consistent of the use of concepts used in the text, for example why do the authors use detriment and not barrier? Sometimes you use social barriers, sometimes socio-ecological facilitators and barriers. The same goes for enablers and facilitators.

In the rational of doing this study you argue that earlier studies of parental perceptions of AST focused primarily on parental intentions and behavioural beliefs, why is this a limit?  In the argumentation of doing this study you also mention the lack of social barriers. In your instrument it seems like you use the term social safety, are these different and then how does instrument contribute to the lack of social barriers? Please provide an explanation for this.

In the aim you use social and physical environment items and in the method section facilitators and barriers? Unify or provide and explanation why.

Response: Thanks for these notes on the consistency of language used in the manuscript. A number of revisions have been to address these points. The terms “detrimental” or “detriment” have now been removed and replaced with the term “obstruction” throughout the revised manuscript.

Regarding the rationale for the study, we have now reframed this argument to be more about comprehensiveness of existing tools and how our using social-ecological theory helps to address some of these current shortcomings: “Moreover, among the current AST-specific scales examining parental perceptions of AST topics, such scales have focused primarily on parental intentions and behavioural beliefs [50] or have lacked the inclusion of a full scope of socio- barrier items [51] in their tools. To support efforts aimed at evaluating school- or community-level AST interventions and develop a tool containing a comprehensive suite of social and physical environment measures, we frame our scale development process within socio-ecological theory.”

Aim:

The aim is already presented in last paragraph? It would be clearer to the reader if the aims were at one place only, using the same adjectives.

Please provide a argument for not using theory in the development phase of the instrument. As this is otherwise what is recommended and really important. If you are not able to provide this explanation, please make this evident in the limitations of the study.

You state in the gap that social barriers are missing in current instruments. But here you only mention barriers not social barriers. In the arguments for this study you claim that other studies do not include social barriers, they do in fact include barriers. In one paper these items are defined as impeding factors and not barriers. Please make this clear.

Response: Thanks for highlighting these inconsistencies. We have revised the end of the introductory paragraph to use the same adjectives as the aims in section 1.3 where these points are detailed in full (“Noting these shortcomings, the aims of this paper are specifically to: i) develop reliable and validated measures delineating parental perceptions of barriers and enablers to AST, and ii) connect these validated measures for the purposes of developing broader constructs for use in the Perceived Active School Travel Enablers and Barriers–Parent (PASTEB–P) questionnaire.”). Regarding the use of theory, we have now included in a new section (i.e., section 1.2) to the introduction outlining how the study was informed by social ecological theory (more details on this are presented in a response below to similar feedback). With respect to the gap, as noted above, we have now rephrased this argument in light of the social-ecological theory-related revisions within the manuscript (i.e., section 1.2).

Method

Data collection: There no such thing as a neutral answer option, because a score on a 3 will be more than 1 right? I don’t think this sentence benefits your paper. However, it is an excellent decision to use 4- point Likert scale. Please revise this.

Response: We agree with this statement. To be clear, while it was recommended by participants to include a ‘neutral’ option for the various measures, in section 2.2. and section 3.1.1 it is explained that these neutral options weren’t included in the final tool.

Table 3: This is a really nice table, making it possible for the reader to follow the development phase, which is hardly ever seen.

Response: Thank you.

Data analysis: In this section the Cronbach Alpha that is stated to be used in the abstract is missing. It would for sure strengthen the paper to add an analysis of the internal consistency. Although we would recommend the authors to use the McDonald Omega for Likert-scales. Please revise this.

Response: Thank you highlighting this. As noted above we have replaced the Cronbach’s Alpha with McDonald’s Omega. These revisions can be found primarily in methods section, but also throughout the revised manuscript.

Confirmatory factor analysis: It is not clear why you use the CFA. The CFA can be used if 1, if there is a clear theory of how the items are supposed to act, because you want to confirm that model or 2, as a second step in the validation process of an instrument that has previously been explored with exploratory factor analysis (EFA)? We think it would strengthen your paper to add sentences supporting this choice. Please provide an explanation for the use of this analysis in the development phase or revise the analysis. If the CFA is done separate for facilitators and barriers we are not sure if it is possible to claim that one instrument has been developed. Rather this is two separate instruments. The authors may need to elaborate on this issue. Please motivate choosing only two goodness of fit indicies. You use AMOS and this will provide more than this goodness of fit indices. Why not report them or decide to a reference the choice made. Please add which version of AMOS.

Response: Thank you for bringing this to our attention. Our study and the development of the questionnaire was informed by social ecological theory, and specifically the socio-ecological model of health behavior, which we have now written into the revised paper. We have explained this theoretical framing with a new section in the introduction (please see section 1.2), touched on this framing in various spots throughout the methods section, and noted the scale within larger conversations of the socio-ecological model of behavior in multiple sections of the revised discussion. We have also added the version of AMOS used “(version 24.0.0)” in our analysis to the revised text in section 2.3.3.

Results

Reliability and Consistency: It is hard as reader to understand why the authors have chosen to dichotomize these items. If the items are going to be used this way, please add this in your paper. If the items are not going to be used this way this is not a part that strengthens your paper, as you risk loosing information in data when dichotomizing and that may be the reason for Kappa values being higher. So that does not in fact improve the stability if your instrument. 

Response: Thanks for highlighting his ambiguity in our writing. In the revised manuscript we have removed the word consistency from this section (i.e., section 3.2.; has also been removed from the corresponding section 2.3.2.). The revised section has been structured to more clearly present that section 3.2 refers to item reliability with no use of the word ‘consistency.’ Dichotomization of the measures, meanwhile, was undertaken to support the utility of tool. Specifically, in order to preserve additional items for usage regarding intervention programming (i.e., those that weren’t included in the confirmatory factor analysis, and thus not used for research purposes), a version of tool featuring these dichotomized measures is presented in Tables 4 and 5 to support intervention programming practice—this is elaborated upon in section 4: “Having developed multiple iterations of the tool, researchers are encouraged to use the final version presented in the supplemental materials, whereas intervention programmers may use the broader range of measures outlined in Tables 4 and 5 for school/community/group surveying.”

Appendix A: It would benefit your paper to redo these figures. These comes directly from the Software, still with names of items used in the software?

Response: Thanks for highlighting this. We have renamed the variables/measures in the photos.

Round 2

Reviewer 3 Report (New Reviewer)

Thank you for answering all my comments and making the required changes in the text. I think this is a well written study that add to existing knowledge. 

This manuscript is a resubmission of an earlier submission. The following is a list of the peer review reports and author responses from that submission.

Round 1

Reviewer 1 Report

The paper 'Validating the Perceived Active School Travel Enablers and Barriers–Parent (Pasteb–P) Questionnaire to Support Intervention Programming and Research' outline and test the construct validity of measures delineating parental perceptions of barriers and facilitators to AST and evaluate the reliability and consistency of the developed measures. I found the manuscript very well-organized and well-written and I suggest publishing the paper. 

Author Response

Thank you for taking the time and effort to review our paper, we appreciate it.

Reviewer 2 Report

Thank you for letting me review this article concerning a very important aspect within health promoting research. The study is very well written and has a high significance and general interest.

I especially appreciate the design using mixed method and participatory approach featuring parents as co-creators situating them as experts in the construct development phase.

I really don´t have any suggestions for improvement other than the suggestions that you might refer to Forsberg et al (2021) “Development and Initial Validation of the PILCAST Questionnaire: Understanding Parents’ Intentions to Let Their Child Cycle or Walk to School” when you describe what has previously been done within the field. 

Author Response

  • Thank you for letting me review this article concerning a very important aspect within health promoting research. The study is very well written and has a high significance and general interest. I especially appreciate the design using mixed method and participatory approach featuring parents as co-creators situating them as experts in the construct development phase.

Response: Thank you for taking the time and effort to review our paper, we appreciate it.

  • I really don´t have any suggestions for improvement other than the suggestions that you might refer to Forsberg et al (2021) “Development and Initial Validation of the PILCAST Questionnaire: Understanding Parents’ Intentions to Let Their Child Cycle or Walk to School” when you describe what has previously been done within the field.

Response: Thank you for bringing this reference to our attention. We have now cited it in our revised manuscript in both the Introduction (please see 1.1 (reference #51), part of the justification for the present paper) and Discussion (please see section 4.1 and 4.3, comparison of current scale to other similar scales) sections of the paper.

Reviewer 3 Report

See document attached.

Author Response

  • This is a very interesting study regarding the development of an instrument to measure active school travel-specific parental perceptions. The literature is well-structured and presents a lot of important information, based on recently published literature. On the other hand, a major concern of mine is about the methodology used, which is not clearly presented in specific parts of the Methods section. In addition, the lack of a confirmatory factor analysis significantly weakens the overall quality of the validation study, since there cannot be a validation study which obviously has factors without an EFA or CFA. More detailed feedback and comments follow:

Abstract

  • “A child’s ability … other considerations”: The sentence is too long; consider revising it and splitting it in 2 sentences.

Response: Thank you for noting this sentence structure issue. We have now broken this sentence into two smaller sentences (please see revised abstract).

  • “At present … decision-making processes”: The sentence, as it is structured, does not make much sense and is difficult to understand. Consider revising it.

Response: Thank you for highlighting this ambiguity. In the revised manuscript we have now edited the sentence to read as follows (in revised abstract): “However, there is currently a lack of AST-specific scales that include validated parental perception measures related to such notable barriers and facilitators, or those which tend to frame their AST decision-making processes.”

Introduction

  • The literature review is adequately developed and thorough and includes current and relevant literature. It adequately introduces the reader to the topic under examination (validation of a tool regarding AST-specific parental perceptions). In addition, the study’s aims and objectives are clearly presented and explained. On the other hand, since this is a validation study, I would suggest the authors to present some information about previous studies that have validated similar instruments in the past. Currently, they have included only 1 sentence in this regard (“Despite the relative importance of parental perceptions of their children’s AST participation, repeated studies have noted methodological concerns regarding the validity of such measures [21-22]”). Information regarding other validation studies and/or instruments are missing from the Introduction section. I believe that this is an important part of this section which is not adequately addressed/explained by the authors.

Response: Thanks for noting this lack of justification on our part. We have made a couple of edits in the revised paper to address this feedback. First, we have given a quick example of one of the relevant methodological concerns that have been previously expressed in this opening paragraph (where the above point is cited, please see section 1): “Despite the relative importance of parental perceptions of their children’s AST participation, repeated studies have noted methodological concerns regarding the validity of such measures [21-22]; for instance, it has been argued that there is a need to further examine factors that parents perceive as facilitating of or detrimental to AST [23].”

Second, we added a two more references that explicate additional justifications for the undertaking of this scale in section 1.1: “As noted above, there is a need to better explore the factors that parents perceive as facilitating of or detrimental to AST and to better conceptualize these factors [23]. Moreover, among the current AST-specific scales examining parental perceptions of AST topics, such scales have focused primarily on parental intentions and behavioural beliefs [50] or have lacked the inclusion of social barrier items [51] in their tools.”

Methods

  • Table 1: “Children living within walking distance”, add “from school”. Also, the authors should explain how they define “walking distance from school” (what does it mean?).

Response: Thanks for highlighting this oversight. We had added a footnote in the table to explain that walking distance was operationalized as <1.6 km.

  • Table 1: Highlight (maybe with bold/italics) the titles “Predominant Mode of Travel to School” and “Predominant Mode of Travel Home from School”.

Response: Thank you for this suggestion, in the revised manuscript we have bolded and italicized these sections of Table 1.

  • The Methods section includes all important information and details that ensure the repeatability of the study. Data analysis: Why did the authors decide to use Kappa Tests to examine repeatability and consistency of the Test-Retest Study survey answers? Usually, in order to test the repeatability of the test-retest results, the Intraclass Correlation Coefficients (ICC) are used. I would recommend the authors to keep the Kappa test analysis, however I would further suggest including the ICC for the test-retest procedure.

Response: Thank you for this suggestion. While we agree that ICC can be used to evaluate repeatability, Kappa test is an acceptable method for measuring test-retest. This is especially the case when measuring matched binary or categorical variables, as is our case (Hallgren, 2012).

Ref: Hallgren, K. A. (2012). Computing inter-rater reliability for observational data: an overview and tutorial. Tutorials in quantitative methods for psychology8(1), 23.

  • From the presentation of the instruments used (as presented in Table 3), it is obvious that the Barriers and Facilitators for AST questions have a factorial structure (with factors such as Skills, Convenience, Road safety, etc.). However, the authors have decided not to include this analysis in the current paper. Face validity is important; however, it is a rather “weak” approach to estimate the validity of a newly developed instrument/scale and this is an important shortcoming of this study. I firmly believe that the statistical analysis should include a factor analysis, and since the structure of the questionnaire is methodologically confirmed through the procedure of face validity, a higher-order confirmatory factor analysis (with 2 higher-order factors; Barriers and Facilitators) should be implemented.

Response: We agree that this would definitely be the next step of this project. The current face validity and test-retest validity analysis of our survey tool as the primary goal of this study. We also believe that the Cronbach’s Alpha does provide conducted for each construct developed for the test and retest datasets are testing for internal consistency that the factors analysis will also be testing. The categories within each metric is driven from the literature, which involved a review of over 100 academic articles that evaluated barriers and facilitators to AST.

We have addressed this weakness in the following ways:

  • We have added clarity around the method for developing the barriers / facilitator questions (in section 2.2): “Table 3 presents the initial set of concepts and their component items that were developed for both studies through our team’s examination of relevant studies evaluating barriers and facilitators to AST, which included a review of over 100 academic articles [e.g., 22,45,52].”
  • Add a limitation regarding the constructs (in section ): “Last, while the component analyses of this paper suggest moderate to high reliability with respect to each of the individual barriers and facilitators that were developed, and thus support their usage with respect to informing STP programs, we caution against the use of individual items, as well as the ‘skills’ (barrier) and ‘supportive environments’ (enabler) concepts, being used as outcome or explanatory variables in larger analyses. Rather the desired use of the developed and validated measures here is ideally in their full format; that is to say, using all items of each developed construct together to determine their aggregated averages in suitable research contexts (e.g., informing STP strategies targeting parental topics). While these constructs were developed through literature informed practice and validated through content validation with parents, future work should verify these constructs through the use of confirmatory factor analysis.”

  • 3.2. Cohen’s* Kappa Test: From this paragraph it is not clear how test-retest reliability of the scale was examined and established. Did the authors actually tested test-retest reliability, which is mentioned previously? If so, how did they test it? Consider using the ICC as mentioned above.

Response: Thanks for pointing out this confusion. We have addressed this by restructuring 2.3.2 to clarify how we did the test. Per our previous response, we have decided to continue using Cohen’s Kappa Test, which is more appropriate for nominal and binary data.

“To address objective 2 (evaluate reliability and consistency of perception measures), Cohen’s* Kappa tests were calculated to determine the level of intra-rater (i.e., within individual) reliability across two time points with the Test-Retest Pilot Study Protocol data (See section 2.2.2 for full details). The Kappa test was conducted for the barriers and facilitators of the test and retest phases using two versions of the data:

  • The original 4-point Likert questions described in Table 3 were assessed with Kappa tests in order to develop more comprehensive scales for AST research (i.e., higher agreement required).
  • Each of the individual items within the different barrier and facilitator concepts were then also recoded into binary responses (i.e., agree vs. disagree) and analyzed with Kappa tests in order to offer simpler and more efficient scales that could be used for intervention development purposes (i.e., lower agreement required).

Kappa results were assessed based on the previously established cut points where ≥0.81 was interpreted as ‘almost perfect’ agreement, 0.61-0.80 was interpreted as ‘substantial’ agreement, and 0.41-0.60 was interpreted as ‘moderate’ agreement [57-59]. This study follows guidelines provided by past work [57-59] that only Kappa values above 0.8 provide sufficient evidence of strong agreement and be able to be used for research evaluating how AST is influenced by individual barriers and facilitators to AST. Barriers and facilitators with lower Kappa values between 0.41 and 0.80 are not as reliable in their agreement and will only be recommended to be used to inform intervention development. All statistical analyses were completed in SPSS (IBM, version 25, Markham, Ontario, Canada).”

  • “After ascertaining the averages for each response item, Cronbach’s Alpha tests were run to assess the internal consistency of each item”. Cronbach a coefficients are usually used to assess the internal validity of a group of items, and not individual items. I am not sure why the authors used this approach to provide evidence of internal consistency. Note: I can see that in the Results section (Table 6, page 13), Cronbach a scores were estimated for the groups of items, and not for every single item. The authors should rephrase this sentence.

Response: Thank you. That was definitely poorly worded. We have clarified as follows:

“Finally, bringing together objectives 2 and 3 (final validation of perception measures into larger constructs for use in the PASTEB–P), Cronbach’s alpha and intraclass correlation (ICC) analyses were run. More pointedly, the measures that were refined in the Validation Pilot Study were examined. To assess individual measures, the average score for each set of questions was calculated using the four-point Likert scale values (i.e., strongly disagree [0]; disagree [1]; agree [2]; strongly agree [3]). After ascertaining the averages for each response item, the results of the literature review and construct validity conducted parents were used to group barriers and facilitators into themed constructs. Cronbach’s Alpha tests were run to assess the internal consistency of each construct [60] wherein alphas between 0.60– 0.74 were interpreted as ‘good,’ while correlations above 0.75 were interpreted as ‘excellent’ [61-63]. After assessing the internal consistency of each construct, ICCs were run to evaluate intra-rater reliability of the results of each construct using data from the Test-Retest Study. ICCs, as an indicator of reliability, are scored on a scale from 0 to 1 with the latter indicating perfect reliability and >0.7 being regarded as ‘substantial’ agreement. All statistical analyses were completed in SPSS.”

  • 3.2. and 2.3.3.: These 2 sections are difficult to understand, and I believe that there are many methodological errors in the way that the statistical analyses are presented. I would advise the authors to present these analyses in a more meaningful and understandable way.

Response: See previous two comments. You are correct. Poor phrasing throughout. Significant editing of these two sections has been completed to clarify the data analysis methods.

  • Since the authors used Cronbach a coefficients to estimate the internal validity of the group of items (e.g., Skills, Convenience, Road safety, etc.), the grouping of the items into factors should have been examined, and the authors should not have relied only on face validity of the scale. This is a serious disadvantage of this study and should be addressed by the authors.

Response: See previous comment. We have added a limitation about this: “Last, while the component analyses of this paper suggest moderate to high reliability with respect to each of the individual barriers and facilitators that were developed, and thus support their usage with respect to informing STP programs, we caution against the use of individual items, as well as the ‘skills’ (barrier) and ‘supportive environments’ (enabler) concepts, being used as outcome or explanatory variables in larger analyses. Rather the desired use of the developed and validated measures here is ideally in their full format; that is to say, using all items of each developed construct together to determine their aggregated averages in suitable research contexts (e.g., informing STP strategies targeting parental topics). While these constructs were developed through literature informed practice and validated through content validation with parents, future work should verify these constructs through the use of confirmatory factor analysis.”

Results

  • “However, in the second analysis where the items were recoded into binary variables of ‘agree’ or ‘disagree’ the Kappa test scores increased considerably”: Since the initial scales had a 4-point Likert type of answering, why did the authors also used binary answers “agree or disagree”? This is not clear from the presentation of the results, and it was not mentioned in the Methods section. Binary answers are usually problematic, as they cannot capture the entire range of possible answers. Kappa test thresholds should have been presented in the Methods section, and not in the results. I would suggest the authors to include this information in the Methods, and provide relevant literature to support the selection of this thresholds (e.g., why 60-80% is considered substantial agreement, and why 81% and above is almost perfect agreement?).

Response: We have added clarification around this discussion in the results as well as references.

“Kappa results were assessed based on the previously established cut points where ≥0.81 was interpreted as ‘almost perfect’ agreement, 0.61-0.80 was interpreted as ‘substantial’ agreement, and 0.41-0.60 was interpreted as ‘moderate’ agreement [57-59].  This study follows guidelines provided by past work [57-59] that only Kappa values above 0.8 provide sufficient evidence of strong agreement and be able to be used for research evaluating how AST is influenced by individual barriers and facilitators to AST. Barriers and facilitators with lower Kappa values between 0.41 and 0.80 are not as reliable in their agreement and will only be recommended to be used to inform intervention development. All statistical analyses were completed in SPSS (IBM, version 25, Markham, Ontario, Canada).”

  • “Results of the analysis examining the agreement among parental responses to perceived facilitators of AST are presented in Table 5”: I would like to ask the authors how they dealt with the items with very low Kappa test coefficients. Did they remove these items from the Cronbach analysis?

Response: Very low kappa tests were not removed from the constructs as we decided that the individual barriers did not have enough validity to be used for research individually. But our hypothesis was that the scales would be able to be combined into a construct and there would be a trend in how parents answered the group of questions. This was tested through Cronbach’s alpha and proved correct for all but skills (which had the lowest kappa values) and Supportive Environments.

Discussion

  • “The results of this study present researchers and intervention evaluators with a validated tool containing …”: I am not sure that this study presents a validated tool. Since factor analysis has not been used to support the construct/factorial validity of an instrument that obviously has factors, this statement is not adequate and lacks specificity. The Discussion does not include any actual discussion of how the authors dealt with items and groups of items (factors) with low scores. Did they retain these items/groups in the questionnaire? Should these items/groups be excluded from future use? This should be discussed.

Response: Referring back to our above responses, and in response to the points noted here, we have now provided several more details in our revised methods sections to enhance clarity regarding the construct validity issues, and added considerable text to the limitations section of the discussion to detail the scope and appropriate use of this work.

Round 2

Reviewer 3 Report

The revised manuscript has been considerably improved.

However, the main limitations highlighted in the previous review still exist and have not been adequately addressed. These include:

1. Use of binary answers, instead of the full possible range of answers, based on the 4-point Likert type scale.

2. Confirmatory Factor analysis not used to support the factorial structure of the scale.

3. The authors did not deal with the items with very low Kappa test coefficients, and these items were retained in the final instrument, even though these should have been removed.

Unfortunately, only addressing these limitations in the Discussion section does not improve the methodological quality of the manuscript, which as it stands write now has many important limitations.

Author Response

Thank you to the reviewer for these questions challenging us to ensure that our methodology is as rigorous as possible. This work has required us to make fairly major changes to our methodology:

  1. We better justified the use of the binary analysis for barriers and facilitators to inform the population intervention.
  2. In reviewing the methods we used, we realized we had used the general Kappa score analysis, but have improved our reliability analysis for the test – retest of individual barriers and facilitators to use the weighted Kappa analysis. This method increases the dependability of the data analysis by accounting for instance of small variance in the results.
  3. We removed any barriers / facilitators from the constructs that had a Kappa score less than 0.4 to ensure that individual barriers with poor reliability were not included in our data analysis.

While we understand the argument the reviewer has for completing the factor analysis, our objective was not to test a complex model assessing latent variables/factors (i.e., theoretical constructs) that are hypothesized to cause our observed variables, nor are we at a point where we are specifying the causal relationship between latent variables. Instead, we have established face validity and using the literature and work of our team, assessed the internal consistency/reliability of the scale items. While we agree that confirmatory factor analysis is possible, we do not think it is necessary for our paper as currently written. We have acknowledged this as a potential limitation in the conclusions.